# A simple model suggesting economically rational sample-size choice drives irreproducibility

**Oliver Braganza**●*

Institute for Experimental Epileptology and Cognition Research, University of Bonn, Bonn, Germany

* oliver.braganza@ukbonn.de

## Abstract

Several systematic studies have suggested that a large fraction of published research is not reproducible. One probable reason for low reproducibility is insufficient sample size, resulting in low power and low positive predictive value. It has been suggested that insufficient sample-size choice is driven by a combination of scientific competition and 'positive publication bias'. Here we formalize this intuition in a simple model, in which scientists choose economically rational sample sizes, balancing the cost of experimentation with income from publication. Specifically, assuming that a scientist's income derives only from 'positive' findings (positive publication bias) and that individual samples cost a fixed amount, allows to leverage basic statistical formulas into an economic optimality prediction. We find that if effects have i) low base probability, ii) small effect size or iii) low grant income per publication, then the rational (economically optimal) sample size is small. Furthermore, for plausible distributions of these parameters we find a robust emergence of a bimodal distribution of obtained statistical power and low overall reproducibility rates, both matching empirical findings. Finally, we explore conditional equivalence testing as a means to align economic incentives with adequate sample sizes. Overall, the model describes a simple mechanism explaining both the prevalence and the persistence of small sample sizes, and is well suited for empirical validation. It proposes economic rationality, or economic pressures, as a principal driver of irreproducibility and suggests strategies to change this.

## Introduction

Systematic attempts at replicating published research have produced disquietingly low reproducibility rates, often below 50% [1–5]. A recent survey suggests that a vast majority of scientists believe we are currently in a 'reproducibility crisis' [6]. While the term 'crisis' is contested [7], the available evidence on reproducibility certainly raises questions. One likely reason for low reproducibility rates is insufficient sample size and resulting low statistical power and positive predictive value [8–12]. In the most prevalent scientific statistical framework, i.e. null-hypothesis-significance-testing (NHST), the statistical *power* of a study is the probability to detect a hypothesized effect with a given sample size. Insufficient *power* reduces the probablity

**Data Availability Statement:** The code to run the model and generate the figures is available a supporting files and can additionally be downloaded at www.proxyeconomics.com.

**Funding:** This study was funded by VW Foundation under the program Originalitätsverdacht, Project: 'Proxyecomomics'. The funder had no role in study design, data collection and analysis, decision to publish, or preparation of the manuscript.

**Competing interests:** The authors have declared that no competing interests exist.

that a given hypothesis can be supported by statistical significance. Insufficient sample sizes therefore directly impair a scientist's purported goal of providing evidence for a hypothesis. Additionally, small sample sizes imply low positive predictive value (*PPV*), i.e. a low probability that a given, statistically significant finding is indeed true [8, 10]. Therefore small sample sizes undermine not only the purported goal of the individual researcher, but also the reliability of the scientific literature in general.

Despite this, there is substantial evidence that chosen sample sizes are overwhelmingly too small [10–15]. For instance in neuroscientific research, systematic evaluation of meta-analyses in various subfields yielded mean power estimates of 8 to 31% [10], substantially less than the generally aspired 80%. Notably, these estimates should be considered optimistic. This is because they are based on effect size estimates from meta-analyses which are in turn likely to be inflated due to publication bias [11, 16]. Remarkably, more prestigious journals appear to contain particularly small sample sizes [11, 17, 18]. Moreover, the scientific practice of choosing insufficient sample sizes appears to be extremely persistent, despite perennial calls for improvement since at least 1962 [11, 13, 19].

Perhaps the most prominent explanation for this phenomenon is the competitive scientific environment [6, 20, 21]. Scientists must maximize the number and impact of publications they produce with scarce resources (time, funding) in order to secure further funding and often, by implication, their job. For instance Smaldino and McElreath have suggested that 'efficient' scientists may 'farm' significant (i.e. publishable) results with low sample sizes [13]. This suggests that sample-size choices may reflect an economic equilibrium or, in other words, that small sample sizes may be economically rational. Notably, economic equilibria may be enforced not only by rational choice but also through competitive selection mechanisms (see Discussion) [13, 22, 23]. The existence of an economic equilibrium of small sample sizes would help to explain both the prevalence and the persistence of underpowering. Recently, this economic argument has been formally explored in two related optimality models [24, 25]. While similar to the present model in spirit and conclusion, these models contain some higher order parameters, creating challenges for empirical validation. Here, we present a simple model, well suited to empirical validation, in which observed sample sizes reflect an economic equilibrium. Scientists choose a sample size to maximize their profit by balancing the cost of experimentation with the income following from successful publications. For simplicity we assume only statistically significant 'positive findings' can be published and converted to funding, reflecting 'positive publication bias'. The model predicts an (economically rational) equilibrium sample size (*ESS*), for a given base probability of true results (*b*), effect size (*d*), and mean grant income per publication (*IF*). We find that i) lower *b* leads to lower *ESS*, ii) greater *d* and *IF* lead to larger *ESS*. For plausible parameter distributions, the model predicts a bi-modal distribution of achieved power and reproducibility rates below 50%, both in line with empirical findings. Finally, we explore the ability of conditional equivalence testing [26] to address these issues and find that it leads to almost uniformly superior outcomes.

## Materials and methods

### Model

Economically rational scientists choose sample sizes to maximize their *Profit* from science given by their *Income* from funding minus the *Cost* of experimentation (Eq 1). For simplicity we assume they receive funding only, if they publish and they can publish only positive results. The first condition reflects the dependence of funding decisions on the publication record as captured by the adage 'publish or perish'. The second condition captures the well documented phenomenon of positive publication bias (see Central Assumptions section below).

Specifically,

$$Profit(s, IF, d, b) = \underbrace{IF \times TPR(s, d, b)}_{Income} - \underbrace{s}_{Cost} \qquad (1)$$

where *IF* is a positive constant reflecting mean grant income per publication (Income Factor), $TPR(s, d, b)$ is the total publishable rate given a sample size ($s$), effect size ($d$) and base probability of true effects ($b$). The latter term ($b$) [27] has also been called the 'pre-study probability of a relationship being true ($R/(R+1)$)' [8]. At the same time scientists incur the cost of experimentation which is assumed to be linearly related to sample size ($s$). For simplicity we scale *IF* as the number of samples purchasable per publication such that the cost of experimentation reduces to $s$. Accordingly, each sample pair costs one monetary (or temporal) unit. $TPR(s, d, b)$ is the sum of false and true positive rates and can be calculated using basic statistical formulas [8, 27] (Eq 2):

$$TPR(s, d, b) = \underbrace{\alpha \times (1 - b)}_{\text{false positive rate}} + \underbrace{(1 - \beta(s)) \times b}_{\text{true positive rate}} \qquad (2)$$

where $\alpha = 0.05$ is the Type-1 and $\beta(s)$ the Type-2 error and $(1 - \beta(s))$ is statistical *power*. The equilibrium sample size (*ESS*) is then the sample size at which *Profit* is maximal.

To model conditional equivalence testing (CET), we assumed the procedure described by [26]. Briefly, all negative results are subjected to an equivalence test, to establish if they are statistically *significant negative* findings. *Significant negative*, here, is defined as an effect within previously determined equivalence bounds ($\pm\Delta$), which are set to the 'smallest effect size of interest' [28]. The total publishable rate for CET ($TPR_{CET}(s, d, b, \Delta)$) is thus the sum of $TPR(s, d, b)$ and subsequently detected *significant negative* findings (Eq 3).

$$TPR_{CET}(s, d, b, \delta) = TPR(s, d, b) + \underbrace{\alpha_{CET} \times b \times \beta}_{\text{false negative rate}} + \underbrace{(1 - \beta_{CET}(s, \delta)) \times (1 - b) \times (1 - \alpha)}_{\text{true negative rate}} \qquad (3)$$

where $\alpha_{CET} = 0.05$ is the Type-1 and $\beta_{CET}(s, \Delta)$ the Type-2 error, $(1 - \beta_{CET}(s, \Delta))$ is statistical *power$_{CET}$* and $\Delta$ is the equivalence bound of the equivalence test. Note the additional correction factors $\beta$ and $(1 - \alpha)$ for the false and true negative rates respectively, which account for the fact that the equivalence test is performed conditionally on the lack of a previous significant positive finding. The power of the CET $(1 - \beta CET(c, \Delta))$ was computed using the two one sided t-tests (TOST) procedure for independent sample t-tests using the TOSTER package in R [29] (TOSTER::power.TOST.two with $\alpha_{CET} = 0.05$). Profit is then computed as above, but with all published findings ($TPR_{CET}(s, d, b, \Delta)$) instead of only positive findings ($TPR(s, d, b)$) contributing to Income.

Positive predictive value (*/mathitPPV*) is computed as the fraction of true published findings to total published findings. All model code is added as supporting information.

## Central assumptions

Our model relies on three central simplifying assumptions, which here shall first be made explicit and justified:

1. Economic equilibrium sample sizes are the result of profit maximization (i.e. optimization).

2. Due to *positive-publication-bias* scientists can publish only positive results, and receive income proportional to their publication rate.

**Table 1. Parameter space, input parameters to the model.**

| Parameter | Description | Range |
|---|---|---|
| base rate (*b*) | base rate of true positive effects (also called pre-study probability of true effect) | 0–1 |
| effect size (*d*) | cohens d (effect normalized to standard deviation) | 0.1–1.5 |
| Income Factor (*IF*) | number of sample pairs purchasable per publication | 100–1000 |

3. Sample size is *chosen* for a set of parameters (*b*, *d*, *IF*, see Table 1) which are externally given (e.g. by the research field).

The first assumption (profit maximization) can most simply be construed as rational choice in the economic sense but may also be the outcome of competitive selection [13]. For instance if funding is stochastic, scientists who choose profit-maximizing sample sizes would have an increased chance of survival. In contexts, where the cost of sampling is mainly researcher time, profit can similarly be interpreted as time. While rational choice would depend on private estimates of the parameters (*b*, *d*, *IF*), competitive selection could operate through a process of cultural evolution, potentially combined with social learning [13, 23]. Importantly, rational choice and competitive selection are not mutually exclusive and potentially cooperative.

The second assumption (positive publication bias), though obviously oversimplified, seems justified as a coarse description of most competitive scientific fields [30]. Even if negative results are published, they may not achieve high impact and translate to funding.

The third assumption (optimization for given *b*, *d*, *IF*) implies that scientists have no agency over the base probability of a hypothesis being true (*b*), the true effect size (*d*) or the mean income following publication (*IF*). Arguably, *b* and *d* are exogenously given by arising hypotheses and true effects while *IF* is likely to be an exogenous property of a research field. Accordingly, a scientific environment with a fixed or constrained combination of the three parameters can be thought of as a scientific niche. Note that this does not preclude the simultaneous occupation of multiple niches by individual scientists, for instance a *high-IF/low-b* niche and a *high-b/low-IF* niche. In combination with the first assumption this implies that scientists choose/ learn/ are selected for specific sample sizes within niches (but may simultaneously occupy multiple niches). Note, that alternative models, in which choice of *b* is endogenized, describe similar results [24, 25].

## Simulation

Simulations were performed in Python3.6 using the StatsModels toolbox [31]. Model code is shown as supporting information. Statistical *power* was calculated assuming independent, equally sized samples (*s*) and a two-sided, unpaired t-test given effect size *d*. Note, that this implies, *IF* should be interpreted as the number of sample pairs purchasable, and *s* indicates the size of one of the samples. We also calculated *power* using a one sample t-test, where *IF* represents the number of individual samples, and all results were robust. The Type-1 error ($\alpha$) is assumed to be 0.05 throughout. Distributions in Fig 3 were generated using the numpy.random module. The input distribution for *d* was generated using a gamma distribution tuned to match empirical findings [11] ($k = 3.5$, $\theta = 0.2$). Input distributions of *b* and *IF* were generated using uniform or beta distributions with $\alpha$ and $\beta$ chosen from $\alpha = (1.1, 10)$, $\beta = (1.1, 10)$ (*IF* values multiplied by 1000). Bimodal distributions were generated by mixing a low and high skewed beta distribution (with the above parameters) with weights (0.5, 0.5, bimodal) or weights (0.9, 0.1, low/ bimodal). From these distributions 1000 values were drawn at random

and the *ESS* computed for each constellation. To compute the implied distribution of emergent *power* and positive predictive values, the corresponding values for each *ESS* were weighted by its *TPR/ESS*. This corrects for the fact that small *ESS* will allow to conduct more studies, but with smaller *TPR*. For instance, a niche with half the *ESS* will allow twice the number of studies, suggesting these studies may be twice as frequent in the literature. However, of these smaller studies, a smaller fraction (*TPR*) will be significant, reducing the relative abundance of these studies in the literature. In fact the two nearly cancel each other, such that the weighting does not significantly affect the emergent distribution.

## Results

### The equilibrium sample size (*ESS*)

We will now first illustrate the basic model behavior with an exemplary parameter set (*b* = 0.2, 0.5; *d* = 0.5; *IF* = 200). The most important model feature is the robust emergence of an economically optimal, i.e. 'rational', equilibrium sample size (*ESS*) at which *Profit*, i.e. the (indirect) *Income* from publications minus the *Cost* of experimentation, is maximal (Fig 1). A scientist's *Income* (Fig 1A, blue & green curves) will be proportional to her publication rate, i.e. her total publishable rate (Eq 2). An optimum sample size (*s*) emerges because this rate must saturate close to the actual rate of true effects (*b*). Specifically, with infinite sample size and resulting infinite *power*, the total publishable rate approaches the rate of actually true effects (*b*) plus a fraction of false positives ($\alpha \times (1 - b)$). The saturation and slope of the *Income* curve will thus depend on *b*, *IF* and *power*, the latter of which is a function of sample size. Conversely, additional samples always cost more, implying that at some sample size additional *Cost* will outpace additional *Income*. Computing *Profit* for increasing sample sizes (Eq 1) therefore reveals an optimal sample size at which *Profit* is maximal, termed the equilibrium sample size (*ESS*; Fig 1B).

At very small sample sizes, insufficient *power* will preclude the detection of true positives, but income from false positives will never fall below $\alpha \times IF$. Increasing sample size can then increase or decrease *Profit*, depending on the resulting increase in statistically significant results. For instance if true effects are scarce ($b \leq 0.2$ for Fig 1), increasing *power* will only modestly increase income (Fig 1A and 1B, blue line), leading to sharply decreasing *Profit*. In

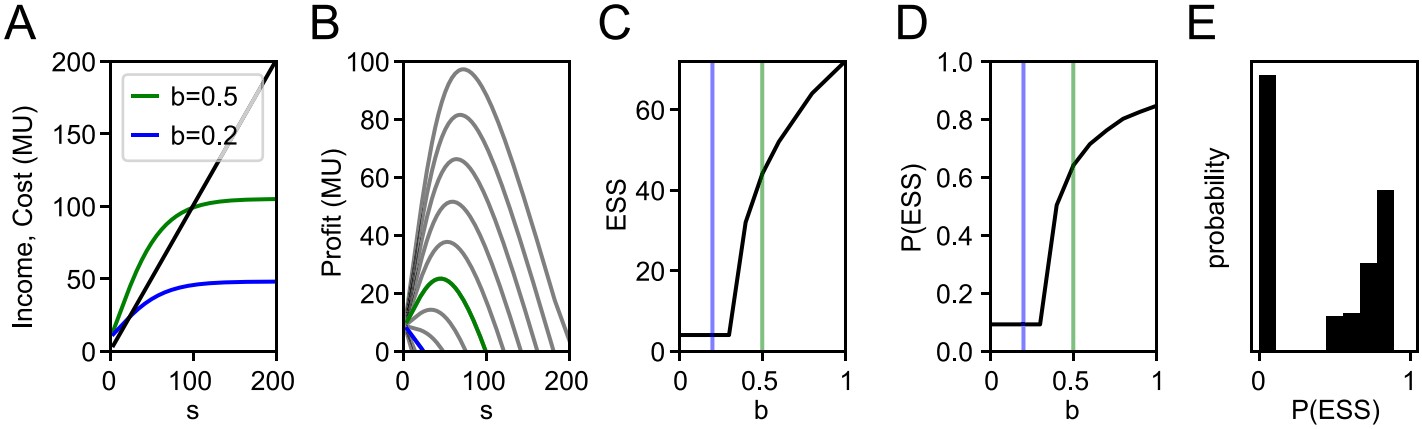

**Fig 1. Equilibrium sample size, basic model behavior illustrated with *d* = 0.5, *IF* = 200. A)** Illustrative *income* (blue, green for *b* = 0.2, 0.5, respectively) and *cost* (black) function with increasing sample size (*s*); MU: monetary units where one MU buys one sample **B)** *Profit* functions for *b* = (0, 0.1,. . ., 1). For any given *b* the (*ESS*) is the sample size at which profit is maximal. **C)** Relation of the *ESS* to *b* (black curve). Respective *ESS* for *b* = 0.2 and 0.5 are indicated by blue and green lines. **D)** Statistical *power* of the *ESS* (P(ESS)) for the given *d* and *IF*. **E)** Expected distribution of statistical *power* at *ESS* if *b* is uniformly distributed.

the extreme (no true effects, $b = 0$) increasing sample size linearly increases *Cost* but the rate of statistically significant (publishable) findings remains constant at $\alpha$. The result is a range of small $b$ at which *ESS* remains at the minimal value ($s = 4$ in our model) (Fig 1C). While we set $s = 4$ as the minimal possible sample size, the minimal publishable sample size may vary by field conventions. The model suggests simply, that there will be an economic pressure toward the *ESS*. This economic pressure should be proportional to the peakedness of the *Profit* curve, i.e. the marginal decrease in *Profit* when slightly deviating from *ESS*. As $b$ increases from zero the peakedness decreases until the *ESS* begins to rapidly shift to larger values (between $b = 0.3$ and 0.4 in our example). At larger values of $b$ peakedness increases again and the *ESS* begins to saturate. Note that adding a constant overhead cost or income per study will not affect the *ESS*. Such an overhead would shift the *Cost* curve (Fig 1A, black) as well as the *Profit* curves (Fig 1B) up or down, without altering the optimal sample size. Accordingly, we find that for a given $d$ and *IF*, hypotheses with smaller base probability, lead to smaller rational sample sizes.

## Statistical *power* at *ESS*

We can now also explore the statistical *power* implied by the *ESS* (Fig 1D). It is most helpful to separate the resultant curve into three phases: i) a range of constant small *power* where *ESS* is minimal, ii) a small range of $b$ ($\approx 0.4 < b < 0.6$) where *power* rises steeply and iii) a range of large $b$ where *ESS* and *power* saturate. Unsurprisingly, where *ESS* is minimal, studies are also severely underpowered. Conversely, where *ESS* begins to saturate, studies become increasingly well powered. Notably, there is only a small range of $b$ where moderately powered studies should emerge. In other words, for most values of $b$, *power* should be either very low or very high. For instance, assuming a uniform distribution of $b$, i.e. scientific environments with all values of $b$ are equally frequent, we should expect a bi-modal distribution of *power* (Fig 1E). If $b$ is already bi-modally distributed, this prediction becomes even stronger. For instance scientific niches may be clustered around novelty driven research with small $b$ and confirmatory research with large $b$. Overall, *power* like *ESS* is positively related to $b$ with a distinctive three phase waveform.

Next, we explored how changing each individual input parameter ($b$, $d$ and *IF*) affected *ESS* and *power* (Fig 2A and 2B, respectively). Specifically, we tested the sensitivity of the $b$ to *ESS* and $b$ to *power* relationships for plausible ranges of $d$ and *IF* (Fig 2A). The *ESS* for a given $b$ should depend both on effect size $d$ (via *power*) and *IF*(via the relative cost of a sample). We reasoned that the majority of scientific research is likely to be conducted in the ranges of $d \in [0.2, 1]$ and $IF \in [50, 500]$ (see Discussion). For small $d$ and *IF* the range of small $b$, where *ESS* and *power* are minimal is expanded (Fig 2A and 2B upper left panels). Conversely, both greater $d$ and *IF* shift the inflection point at which larger sample sizes become profitable rightward. Accordingly, in this domain, above minimal sample sizes should be chosen even with small $b$ (Fig 2A, lower right panels). When $d$ and/or *IF* are large enough, *ESS* leads to well powered studies across most values of $b$ (Fig 2B, lower right panels). Accordingly the distinctive three-phase waveform is conserved throughout much of the plausible parameter space but breaks down towards its edges. These data suggest that, ceteris paribus, a policy to increase any of the three input parameters, individually or in combination, will tend to promote better powered studies. For instance, increasing the funding ratio and therewith *IF* or funding more confirmatory research with high $b$, should both lead to better powered research.

## Emergent *power* distributions for plausible input parameter distributions

In real scientific settings a number of distributions of effect sizes $d$, income factors *IF* and base probabilities $b$ are plausible. We therefore next investigated the emergent *power* distributions for multiple distributions of input parameters (Fig 3). We then calculated the *ESS* and resultant

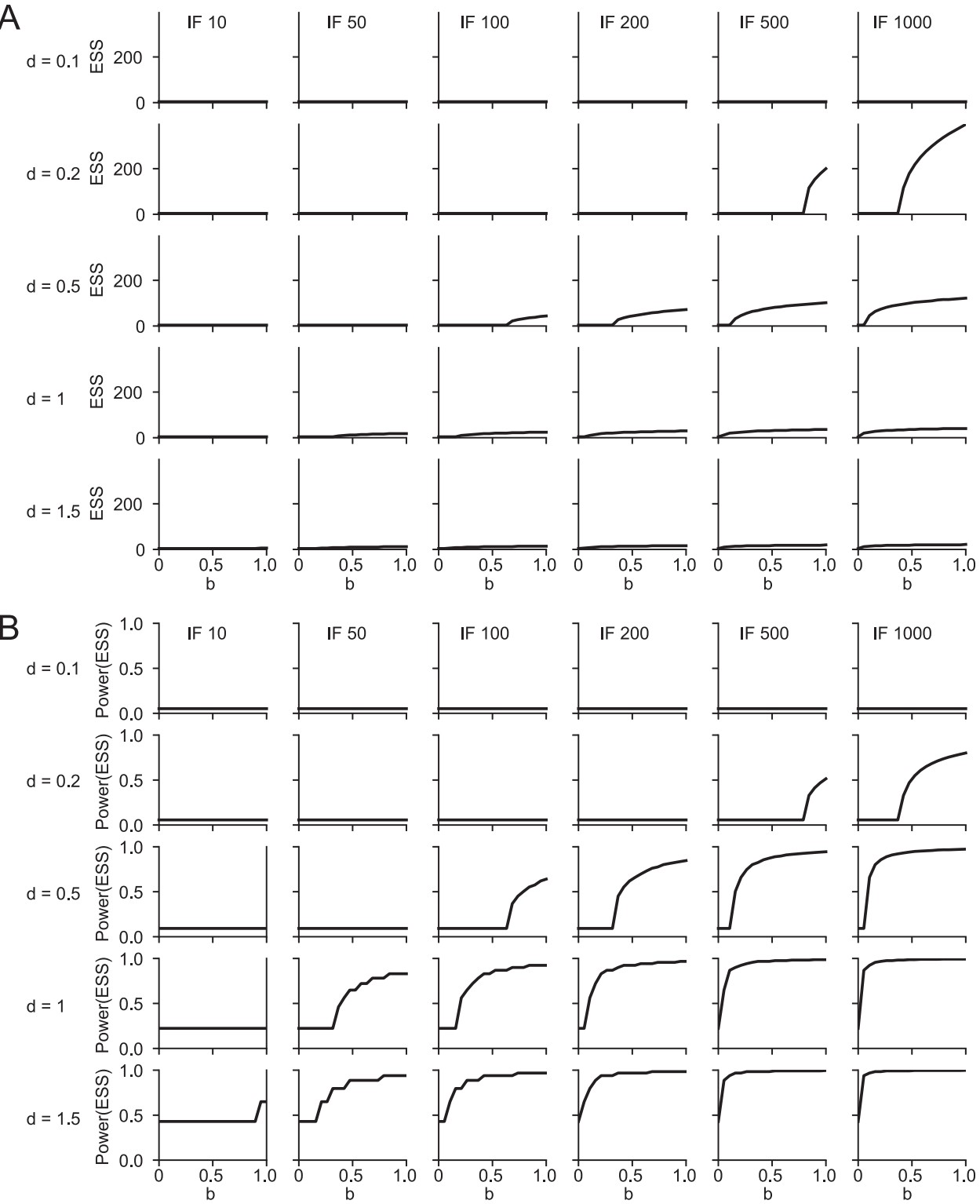

**Fig 2.** Effect of *d* and *IF* on *ESS*, **A)** Each individual line depicts the *ESS* as a function of *b* for a given combination of *d* and *IF*. **B)** Statistical *power* resultant from the *ESS* in panel (A).

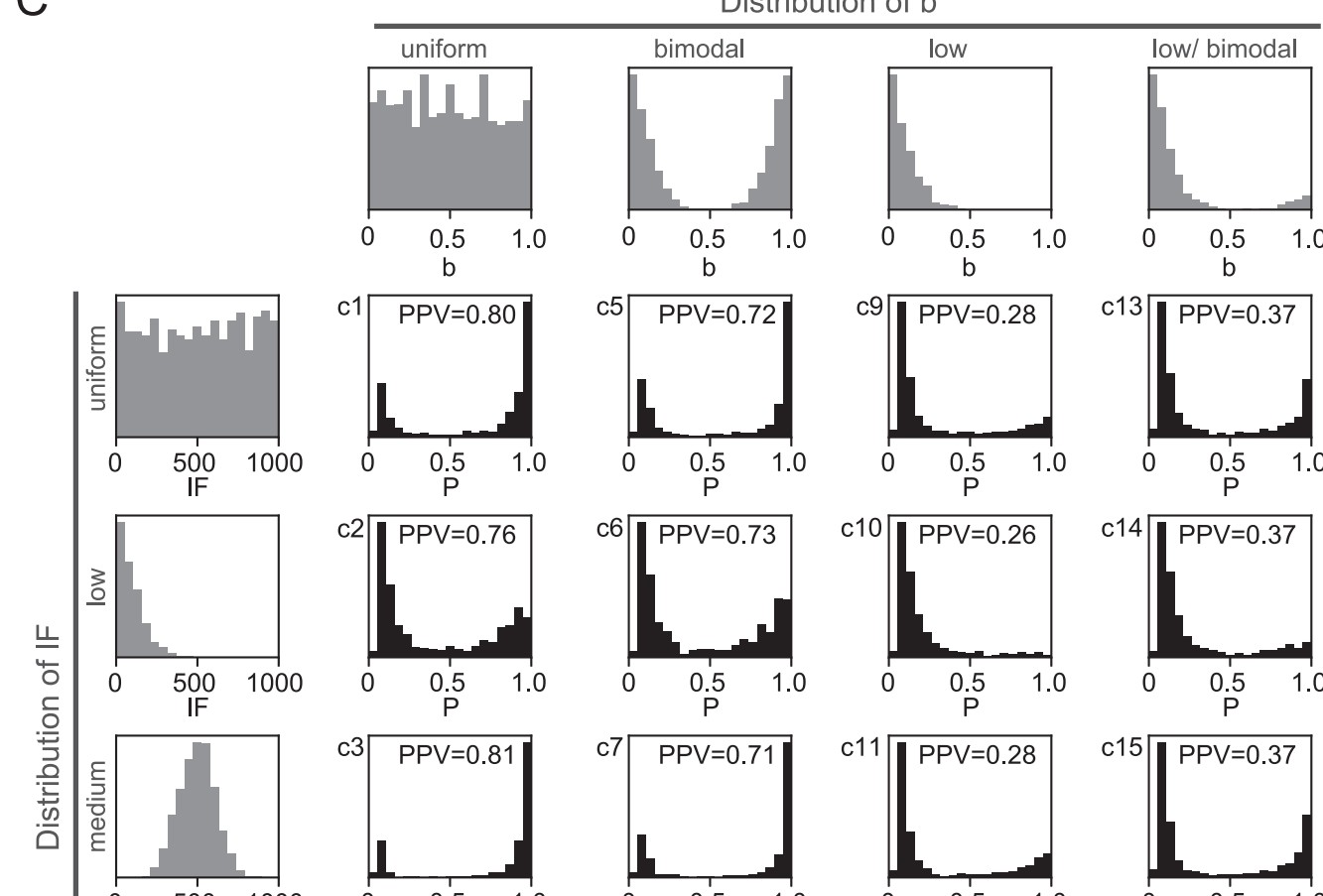

**Fig 3. Distributions of statistical *power* for plausible input parameter distributions, random input parameter constellations were drawn from a range of plausible, simulated distributions (grey, see Methods).** For each input parameter constellation the resultant *ESS* and *power* were calculated. **A)** Summary of model in- and outputs and the probed distributions. **B)** Empirically matched distribution of effect sizes *d* [11], used for all output distribution in C. **C)** Emergent *power* distributions and mean positive predictive values for each combination of distributions of *b* and *IF*.

*power* for each random input parameter constellation (Fig 3A). The distribution of effect sizes was modeled after empirical data [11] with the majority of effects in the medium to large range (Fig 3B, see Discussion). Since the true distribution of *IF* is unknown, we modeled a range of distributions below *IF* = 1000. We reasoned that a single publication leading to funding for 1000 sample pairs was a conservative upper bound in view of published sample sizes [11]. Within this range we probed a uniform distribution as well as low, medium and high distributions of *IF*. Note, that these data also demonstrate the predicted consequences of increasing or decreasing *IF*. Since the true distribution of *b* is similarly unknown, we first probed the uniform distribution (minimal assumption, $Fig\ 3C_{1-4}$) and a bimodal distribution (assuming a cluster of exploratory fields with low *b* and confirmatory fields with high *b*, $Fig\ 3C_{5-11}$). In both these cases the mean *b* is by definition around 0.5, i.e. as many hypotheses are true as are false. However, many scientific areas place an emphasis on 'novelty' suggesting substantially lower *b* [8, 13, 32]. We therefore also probed two more realistic distributions of *b*, namely low (most values around 0.1, $Fig\ 3C_{9-12}$) and low/ bimodal (low mixed with a minor second mode with high *b* $Fig\ 3C_{13-16}$). The latter models a situation where most studies (90%) are exploratory and the remaining studies are confirmatory. We found the resulting bimodal distribution of *power* to be robust throughout, with only the relative weights of the peaks changing. Next we investigated the mean reproducibility rates which could be expected for the resultant distributions. The positive predictive value (*PPV*) measures the probability that a positive finding is indeed true. It thus provides an upper bound on expected reproducibility rates (as the *power* of reproduction studies approaches 100%, reproducibility rates will approach *PPV*). Note that the *PPV* should be interpreted in light of the underlying *b*. For instance, for the first two distributions of *b* ($Fig\ 3C_{1-11}$), the mean base probability is already 50%, so *PPV* < 0.5 would indicate performance worse than chance. For more realistic distributions of *b* ($Fig\ 3C_{12-16}$) *PPV* ranged from 0.26 ($Fig\ 3C_{10}$) to 0.4 ($Fig\ 3C_{16}$), comparable to reported reproducibility rates. Thus, for plausible parameter distributions, rational sample-size choice robustly leads to a bimodal distribution of statistical *power* and expected reproducibility rates below 50%. Additionally, these simulations suggest that creating more research environments with high *b* (e.g. $Fig\ 3C_{5-8}$), for instance in the form of research institutions dedicated to confirmatory research [33], should lead to larger sample sizes and higher reproducibility. Finally, more scientific environments with large income per publication (*IF*), for instance through higher funding ratios, should lead to better powered science and higher reproducibility rates.

## Conditional equivalence testing (CET)

An additional potential strategy to address the economic pressure towards small sample sizes is conditional equivalence testing (CET) [26, 29] (Fig 4). In CET, when a scientist fails to find a significant positive result in the standard NHST, she continues to test if her data statistically support a null effect (*significant negative*). A *significant negative* is defined as an effect within previously determined equivalence bounds (±Δ), which are set to the 'smallest effect size of interest' [28]. This addresses one of the main (and legitimate) drivers of positive publication bias, namely that absence of (significant) evidence is not evidence of absence [34]. Assuming that CET thus allows the publication of statistically *significant negative* findings in addition to significant positive findings implies i) an increase in the fraction of research that is published, ii) a resulting additional source of income from publication without additional sampling cost and iii) an additional incentive for sufficient statistical power, as we will see below.

A crucial step in CET is the *a priori* definition of an equivalence bound (±Δ), below which effects would be deemed consistent with a null-effect. While this can be conceptually challenging in practice [28], it is important to point out that considering a 'smallest effect size of

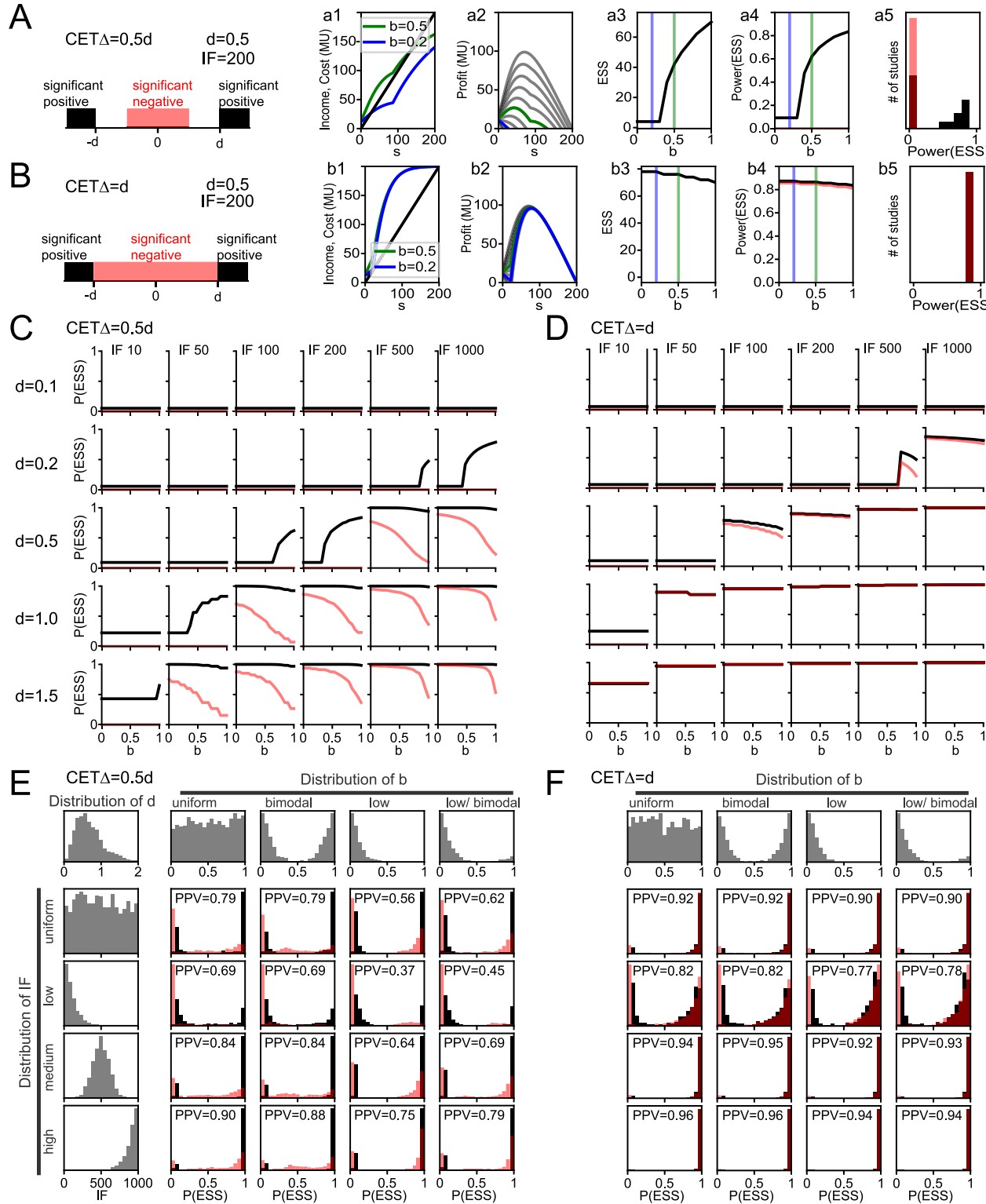

**Fig 4. Conditional equivalence testing.** Exploration of model behavior under conditional equivalence testing. **A)** Basic model behavior illustrated with $d = 0.5$, $IF = 200$, $\Delta = 0.5d$. Left: illustrations of effect sizes that would be considered significant postives (black) or negatives (red). Subpanels show emergent *income* and *cost* curves (a1), resulting *Profit* (a2), resulting $ESS_{CET}$ (a3), resulting *power* in black and $power_{CET}$ in red (a4), and resulting power distribution given uniformly distributed $b$ (a5). For details see Fig 1. **B)** same as A but for $\Delta = d$. **C)** Statistical *power* in black and $power_{CET}$ in red at $ESS_{CET}$ for $\Delta = 0.5d$ analogous to (a4) for various input parameter constellations of $d$ and $IF$. For details see Fig 2. **D)** Same as C, but for $\Delta = d$ analogous to (b4). **E)** Distributions of emergent statistical *power* in black and $power_{CET}$ in red for plausible input parameter distributions given $\Delta = 0.5d$. For details see Fig 3. **F)** Same as E but for $\Delta = d$.

interest' is often suggested as an implicit justification for small sample sizes. In the following, we explore the effects of CET given either $\Delta = 0.5d$ or $\Delta = d$ and $\alpha = 0.05$ for both NHST and CET (Fig 4). Defining $\Delta$ in terms of $d$ allows us to consider the same range of scientific environments, with variable expectations of $d$ as in the previous analyses (Figs 2 and 3). If $\Delta = 0.5d$, a scientist is interested in detecting an effect of size $d$ as above, but is somewhat uncertain how smaller effect sizes ($0.5d$ to $d$) should be interpreted (Fig 4A). However, if her data support an effect $<0.5d$ she would interpret her finding as a significant negative finding and publish it as such. The power of the CET ($power_{CET}$) to detect a negative effect in this setting is substantially smaller than the power of the original NHST $power$, leading to a second shoulder in the income curve at higher sample sizes (Fig 4a1). For the shown example ($\Delta = 0.5d$, $d = 0.5$, $IF = 200$), this does not affect the $ESS$ (Fig 4a2 and 4a3) or the resulting $power$ to detect a positive effect (Fig 4a4 and 4a5, black). Note that each $ESS$ now implies not only a $power$ to detect positive effect but also a $power_{CET}$ to detect a negative effect (Fig 4a4 and 4a5, black and red, respectively), and that even $ESS$ with adequate $power$ can have low $power_{CET}$. If $\Delta = d$ (Fig 4B), the same procedure is applied by the scientists, but all effects statistically significantly smaller than $d$ will be published as negative findings. Note that this does not imply all studies are published, since studies with insufficient power to detect either positive or negative findings will remain inconclusive. Ceteris paribus and $\Delta = d$, $power_{CET}$ is comparable to the $power$ of the original NHST. This has two consequences: i) the income shoulders of the two tests align, boosting profit at the respective sample size and ii) the dependency of Profit on $b$ was largely removed (Fig 4b1 and 4b2). Indeed, the added economic incentive to detect and report significant negative findings led to an inversed relationship of $b$ to $ESS$, since more frequent true negatives (low $b$) implies higher potential profits from adequate sample sizes for the CET.

Systematically probing the input parameter space (as in Fig 2B) showed that CET did not remove the general dependencies of $power$ on $IF$ or $d$ (Fig 4C and 4D). However, the boundary region of $IF$ and $d$ where adequate power first becomes economical shifted toward smaller values, particularly for small $b$. For instance, for $IF = 100$, $d = 1$ and $b \leq 0.2$, the $power$ at $ESS$ shifts from $\approx 20\%$ to $\approx 100\%$. This occurred for $\Delta = 0.5d$ (Fig 4C) and even more prominently for $\Delta = d$ (Fig 4D), where the relation of $b$ to $power$ is almost completely removed.

Finally, we probed the effect of CET on various input parameter distributions, analogous to Fig 3 (Fig 4E and 4F). We find that CET with $\Delta = 0.5d$ leads to improved $power$ and $PPV$ for most parameter distributions, but particularly for more realistic ones (Fig 4E, low and low/bimodal distributions of $b$). This is even more true when $\Delta = d$ where $PPV$ was at 90% or higher for all but the low $IF$ distribution. These results suggest that CET could be a useful tool to change the economics of sample size, increasing not only the publication rate of negative findings but also mean statistical power and thereby the reproducibility of positive findings.

## Discussion

Here, we describe a simple model in which sample-size choice is viewed as the result of competitive economic pressures rather than scientific deliberations (similar to [24, 25]). The model formalizes the economic optimality of small sample size for a large range of empirically plausible parameters with minimal assumptions. Additionally, it makes several empirically testable predictions, including a bimodal distribution of observed statistical power. Given the simplicity of the model, the apparent similarity between its predictions and empirically observed patterns is remarkable. Finally, our model allows to explore a range of policy prescriptions to address insufficient sample sizes and irreproducibility. The core model suggests any policy that increases mean funding per publication or the rate of confirmatory research should lead to better powered studies and increased reproducibility. Additionally, conditional equivalence

testing may address publication bias and provide an economic incentive for better powered science.

## Model predictions and empirical evidence

Our core model predicts i) a correlation between base probability and sample size, ii) a correlation between effect size and sample size, iii) a correlation between mean grant income per publication and sample size. Moreover, for plausible parameter distributions the model predicts iv) a bimodal distribution of achieved statistical *power* and v) low overall reproducibility rates. For the purpose of discussion, it may be of particular interest to contrast our predictions, based on economically driven sample-size choice, to predictions derived from presumed scientifically driven sample-size choices. For instance scientifically driven sample-size choice might be expected to i) require larger samples for more unlikely findings, ii) require larger samples for smaller effects and iii) be independent of grant income. Moreover, scientifically driven sample sizes might be expected to iv) lead to a unimodal distribution of *power* around 80% and v) imply *PPVs* and reproducibility rates above 50%. Which sample sizes are scientifically ideal is of course a complex question in itself, and will depend not only on the cost of sampling but also on the scientific values of true and false positives as well as true and false negatives. Miller and Ulrich, [35] present a scientifically normative model of sample size choice, formalizing many of the above intuitions (however, importantly they do not account for the possibility that negative findings may not enter the published literature). Overall, a prevalence of underpowered research certainly leads to a range of problems, ranging from low reproducibility rates to unreliable metaanalytic effect size estimates [36]. Accordingly, the currently available empirical evidence appears more in line with the economically normative than the scientifically normative account.

ad i) The available evidence suggests that journals with high impact and purportedly more novel (low *b*) findings feature smaller sample sizes [11, 17, 18], in line with our prediction. This finding seems particularly puzzling given the increased editorial and scientific scrutiny such 'high-impact' publications receive. Accordingly, publication in a 'high-impact' journal is generally considered a signal of quality and credibility [37]. From the perspective of an individual scientist, increasing sample size strictly increases the probability of being able to support her hypothesis (if she believes it is true) but does not alter the probability of rejecting it (if she expects it to be false). Similarly, a scientific optimality model assuming true and false positive publications have equal but opposite scientific value, suggests more unlikely findings merit larger power [35]. All these considerations suggest high impact journals should contain larger sample sizes, highlighting a need for explanation.

ad ii) The available evidence suggests a negative correlation between effect size and sample size, seemingly contradicting our prediction [15, 17, 38]. However, the authors caution in the interpretation of this result due to the winner's curse phenomenon [10, 17, 39]. This well documented phenomenon produces an negative correlation between sample size and estimated effect size even when a single hypothesis (i.e. single true effect size) is probed in multiple independent studies. It arises, because for small sample sizes only spuriously inflated effect sizes become statistically significant and enter the literature (also due to positive publication bias). Relating effect sizes from meta-analyses to original sample sizes by scientific subdiscipline may help to overcome this confound.

ad iii) We are unaware of evidence directly relating mean grant income per study to sample size. A study by Fortin and Currie [40] suggests diminishing returns in total impact for increasing awarded grant size. However, impact was assessed without reference to sample sizes. Furthermore, awarded grant size does not necessarily reflect mean grant income, since

larger grants may be more competitive. Indeed, more competitive funding systems are likely to a) increase the underlying economic pressures and b) have additional adverse effects [41]. Indirect evidence in line with our prediction is presented by Sassenberg and Ditrich (2019) [42]. The authors show that larger sample sizes were associated with lower costs per sample. Since *IF* is expressed as 'samples purchasable per publication' this is analogous to higher *IF* correlating with greater sample size.

ad iv) The prediction of a bimodal distribution of *power* is well corroborated by evidence [10, 14, 15]. Particularly the lack of a mode around 80% *power* in our model, as well as all empirical studies is notable. By comparison a scientific value driven model by Miller and Ulrich [35] suggests a single broad mode at intermediate levels of power.

ad v) The predicted low overall reproducibility rates are in line with empirical data for many fields. Two, now prominent, studies from the pharmaceutical industry suggested reproducibility rates of 11 and 22% [1, 2]. Academic studies from psychology and experimental economics found 36, 61 and 62% [3–5]. Our results suggest that differences in these numbers may be driven, for instance, by different base probabilities of hypotheses being true in the different fields.

The present model thus helps to explain a range of empirical phenomena and is amenable to closer empirical scrutiny in the future. Crucially, all in- and output parameters are in principle empirically verifiable. Future studies could, for instance, fit observed *power* distributions with the present model versus alternative formal models. This could directly generate predictions concerning input parameters which could in turn be empirically tested.

## Niche optimization through competitive selection

A central assumption of this model is that sample size is optimized for a given set of parameters (*b*, *d*, *IF*). One way to interpret this is that scientists make rational sample size choices based on their estimates of these parameters for each hypothesis. As noted above, maximizing profit in this context need not be an end in itself but can also be seen as a strategy to secure scientific survival, given the uncertainty of both the scientific process as well as funding decisions. Alternatively, optimization may occur by selection mechanisms, where sample sizes are determined through a process of cultural evolution [13]. In this case one must however make the additional assumption, that parameters remain relatively constant within the scientific niche in which sample sizes are selected. In such a case researchers must only associate the scientific niche with a convention of sample size choice. These conventions could then undergo independent evolution in each niche. Such scientific niches may correspond to scientific subdisciplines and may indeed be identifiable on the basis of empirically consistent sample sizes. We did not address how scientists should distribute their efforts into multiple niches (e.g. exploratory research and confirmatory research). This question has been previously addressed in a related optimality model [24]. The authors suggest that, given prevailing incentives emphasizing novel research, the majority of efforts should be invested into research with low *b*. This is reflected in the present model by the low skewed distributions of *b* (Fig 3C$_{9-16}$). Future evolutionary models could further investigate how mixed strategies of sample size choice perform when individual parameters vary within niches, or scientists are uncertain of the niche.

## Input parameter range estimates

We probed the arising *ESS* for what we judged to be plausible ranges of the three input parameters (*b*, *d*, *IF*):

The base (or pre study) probability of true results is often assumed to be small ($b < 0.1$) for most fields [13, 32]. This is in part because of a focus on novel research [24]. At the same time

there is a small fraction of confirmatory studies, where substantial prior evidence indicates the hypothesis should be true, and *b* should thus be large. We therefore chose to cover the full range of *b* ($b \in [0, 1]$) in addition to some plausible distributions. In light of the considerations by [13] and [24], our distributions might be judged conservative in that real values of *b* may be lower. Notably, models endogenizing choice over *b* reach similar conclusions [24, 25].

We probed a plausible range ($d \in [0.1, 1.5]$) and an empirically matched distribution of *d* [11]. By comparison, in psychological research frequently cited reference points for small, medium and large effect sizes are *d* = 0.2, 0.5 and 0.8, respectively.

Notably our empirically matched distribution (Fig 3B) is based on published effect sizes, which are likely exaggerated due to the winner's curse [17]. For instance [3, 4] find that true effect sizes are on average only around 50 to 60% of originally published effect sizes. This again renders our estimates conservative, in that true values may be lower.

An empirical estimate of *IF* is perhaps most difficult, since the full cost per sample (time, wage, money) may be difficult to separate from other arising costs. Note that a constant overhead cost or income, which is independent of sample size, should not alter the optimal sample size. Nevertheless, we reasoned that plausible values for *IF* should be somewhere within the range of 10 to 1000. For instance, a typically reported sample size is 20 [10, 11]. *IF* must allow to cover the cost of the positive result plus however many unpublished additional samples were required to obtain it. An *IF* of 1000 would thus allow for up to 49 negative findings (or 980 unpublished samples). Given that science does not seem to provide substantial net profits, larger values for *IF* seem implausible. Note, that our model assumes linearly increasing cost for increasing sample size. We found that, for the curvature of the cost function to play a major role for *ESS*, it would need to be very prominent around the range of sample sizes where the curvature of the power-function is strongest. For simple non-linear functions such as a cost exponent between 0.5 and 1.1, we found model behavior to be similarly captured by a linear cost function with adjusted slope. However, strongly non-linear cost functions might affect *ESS* beyond the effect of slope (*IF*).

Together, these considerations suggest that our predictions of *power* and expected reproducibility rates are more likely to be over- than under estimated. Of course, the forces affecting real sample size choices are (hopefully) not solely the economic ones investigated here. Specifically, scientific deliberations about appropriate sample size should at least play a partial role. Indeed, the model predicts the minimal sample size over a wide range of parameters. This could for instance be the minimal sample size accepted by statistical software. Above, we have suggested it is the minimal sample size deemed acceptable in a scientific discipline. This implies that discipline specific norms on minimal acceptable sample sizes reflect scientific deliberations and are enforced during the editorial and review process. Such non-economic forces may be particularly relevant where the *profit* peak is broad.

## Reproducibility

In our model low reproducibility rates appear purely as a result of the economic pressures on sample size. Many additional practices, such as p-hacking, may increase the false positive rate for a given sample size [43–47]. The economic pressures underlying our model must be expected to also promote such practices. Moreover, many of these practices are likely to become more relevant for many small studies. For instance flexible data analysis will increase the probability of false positives for each study. Moreover, in small studies substantial changes of effect size may result from minor changes in analysis (e.g. post-hoc exclusion of data points), thus increasing the relative power of biases. While substantial and mostly laudable efforts are

being made to reduce such practices [43, 48], our approach emphasizes that scientists may in fact have limited agency over sample size, given the economic constraints of scientific competition. Given these constraints, our model suggests reproducibility can be enhanced by policies that i) increase the fraction of research with higher $b$ (e.g. more confirmatory research), ii) lead to higher $IF$ (e.g. higher funding rates), or iii) via the introduction of conditional equivalence testing (CET) [26]. Several previous related models have explored the effect of various policy prescriptions in the light of such economic constraints [24, 25]. For instance Campbell and Gustafson [25] explore the effects of increasing requirements for statistical stringency (e.g. setting $\alpha = 0.005$) as called for by a highly publicized recent proposal [49]. However, they find that this may dramatically lower publication rates, effectively increasing waste (unpublished research) and competitive pressure as well as reducing the rate of 'breakthrough findings'. While we did not perform an extensive investigation of this policy, our model confirms that with ($\alpha = 0.005$), a large fraction of scientific niches, particularly with small ($b$), start yielding net losses. Alternative proposals which directly adress the underlying economic pressures are very much in line with our results [26, 33]. Campbell and Gustafson (2018) propose conditional equivalence testing to increase the publication value of negative findings. Indeed, incorporating their procedure into our model, showed that CET should not only address publication bias, by allowing the publication of more negative findings, but also lead to improved power and reproduciblity of positive findings. In practice, the definition of the equivalence bounds as well as the publication and factual monetary rewarding of negative findings, may render the adoption of CET difficult. However, it is important to note, that even a partial adoption should promote the predicted benefits. Moreover, these hurdles can also be addressed by funding policy. More ambitiously, Utzerath and Fernandez [33] propose to complement the current 'discovery oriented' system with an independent confirmatory branch of science, in which secure funding allows scientists to assess hypotheses impartially. Indeed, such a system may have many positive side effects. A consistent prospect of replication may act as an incentive toward good research practices of discovery oriented scientists. An increased number of permanent scientific positions might reduce the pressure toward bad research practices. Additionally, the emergence of a body of confirmatory research would allow to address many pressing meta-scientific questions, including biases in meta-analytic effect size estimates [36], actual frequencies of true hypotheses [32] as well as true reproducibility rates of published literature [5, 8].

## Relation to proxyeconomics

Our model is consistent with, and an individual instance of, proxyeconomics [23]. Proxyeconomics refers to any competitive societal system in which an abstract goal (here: scientific progress) is promoted using competition based on proxy measures (here: publications). In such cases, the measures or system may become corrupted due to an overoptimization toward the proxy measure [50–54]. As discussed above, such systems have the general potential to create a situation of limited individual agency and system-level lock-in [23]. The present model shows how the specific informational deficits of a proxy allow to create pattern predictions of the potentially emergent corruption. Specifically, an informational idiosyncrasy of the proxy (positive publication bias) leads to a number of predictions which can be i) empirically verified, ii) contrasted with alternative models (see above), and iii) leveraged into policy prescriptions. A similar pattern prediction derived from positive publication bias is the winner's curse [17]. Together, such pattern predictions provide concrete and compelling evidence for competition induced corruption of proxy measures in competitive societal systems.

## Conclusion

Our model strengthens the argument that economic pressures may be a principle driver of insufficient sample sizes and irreproducibility. The underlying mechanism hinges on the combination of positive publication bias and competitive funding. Accordingly, any policy to address irreproducibility should explicity account for the arising economic forces or seek to change them [25, 33].

## Supporting information

**S1 File. Model code for quick reference.** Code to compute the *ESS* (and associated parameters) based on *b*, *d*, *IF*.
(PDF)

**S2 File. Model code for quick reference.** Code to compute the $ESS_{CET}$ (and associated parameters) based on *b*, *d*, *IF*, Δ.
(PDF)

**S3 File. Full python code to compute the *ESS* (and associated parameters) based on *b*, *d*, *IF* and to generate all figures.**
(PY)

**S4 File. Full python code to compute the $ESS_{CET}$ (and associated parameters) based on *b*, *d*, *IF*, Δ and to generate all figures.**
(PY)

## Acknowledgments

Special thanks to Heinz Beck for making this project possible, to Jonathan Ewell for comments on the manuscript and to Harlan Campbell pointing me towards conditional equivalence testing.

## Author Contributions

**Conceptualization:** Oliver Braganza.

**Formal analysis:** Oliver Braganza.

**Funding acquisition:** Oliver Braganza.

**Software:** Oliver Braganza.

**Validation:** Oliver Braganza.

**Visualization:** Oliver Braganza.

**Writing – original draft:** Oliver Braganza.

**Writing – review & editing:** Oliver Braganza.

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
