## [Decision Letter · Decision Letter 0]

11 Dec 2019

PONE-D-19-26742

Economically rational sample-size choice and irreproducibility

PLOS ONE

Dear Dr. Braganza,

I write you in regards to the manuscript PONE-D-19-26742 entitled “Economically rational sample-size choice and irreproducibility” which you submitted to PLOS ONE.

I have solicited advice from two expert Reviewers, who have returned the reports shown below. The two reviewers provide positive recommendations, but both agree that the paper would benefit from a round of revisions before acceptance.

Reviewer #1 suggests that “the manuscript would be stronger if more specific, concrete, prospective predictions were made” and that “it would have been interesting to explore the impact of modifying the input parameters on the predicted outputs of the model”. I encourage you to explore some of the modifications proposed by the reviewer.

Reviewer #2 also suggests going beyond the analysis provided in the current version of the paper and exploring some modifications of the model. Additionally, (s)he raises a number of minor issues that could strengthen the paper.

Based on the Reviewers' reports and my own reading of the paper, I came to the decision to offer you the opportunity to revise the manuscript. If you decide to prepare a substantially revised version of the paper, please provide a detailed response to both Reviewers regarding how you have addressed their concerns. If you resubmit, I would ask the same two Reviewers to review again the paper.

We would appreciate receiving your revised manuscript by Jan 25 2020 11:59PM. To enhance the reproducibility of your results, we recommend that if applicable you deposit your laboratory protocols in protocols.io, where a protocol can be assigned its own identifier (DOI) such that it can be cited independently in the future. For instructions see: http://journals.plos.org/plosone/s/submission-guidelines#loc-laboratory-protocols

We look forward to receiving your revised manuscript.

Kind regards,

Luis M. Miller, Ph.D.

Academic Editor

PLOS ONE

2. Please consider changing the title so as to meet our title format requirement (https://journals.plos.org/plosone/s/submission-guidelines). In particular, the title should be "Specific, descriptive, concise, and comprehensible to readers outside the field" and in this case it is not informative and specific about your study's scope and methodology.

3. Please do not include funding sources in the Acknowledgments or anywhere else in the manuscript file. Funding information should only be entered in the financial disclosure section of the submission system. https://journals.plos.org/plosone/s/submission-guidelines#loc-acknowledgments

4. Please upload a copy of Figure 4, to which you refer in your text on page 6. If the figure is no longer to be included as part of the submission please remove all reference to it within the text.

Reviewers' comments:

Reviewer's Responses to Questions

**Comments to the Author**

1. Is the manuscript technically sound, and do the data support the conclusions?

Reviewer #1: Yes

Reviewer #2: Yes

2. Has the statistical analysis been performed appropriately and rigorously? 

Reviewer #1: Yes

Reviewer #2: Yes

3. Have the authors made all data underlying the findings in their manuscript fully available?

Reviewer #1: Yes

Reviewer #2: Yes

4. Is the manuscript presented in an intelligible fashion and written in standard English?

Reviewer #1: Yes

Reviewer #2: Yes

5. Review Comments to the Author

Reviewer #1: The author presents a relatively simple model, whereby scientists choose “economically rational” sample sizes on the basis of the trade off between the costs of collecting data (with larger studies being more expensive) and the value of publications (i.e., resulting grant income). The authors acknowledge the relatively simplicity of the model, but suggest that the model performs reasonably well in terms of predicting the distribution of statistical power in a way that mirror empirical efforts to estimate this.

The work is a valuable addition to the literature, and complements previous efforts that have taken a similar approach (i.e., using modelling) to understand the impact of incentive structures on the behaviour of scientists, and the resulting quality of published research. However, I felt that (notwithstanding the virtues of simplicity) it would have been interesting to explore the impact of modifying the input parameters on the predicted outputs of the model (i.e., on scientists’ behaviour).

For example, the drivers of small sample size in the model are low base probability, small effect size, and/or low grant income per publication. One could argue that at least two of these are shaped by funding agencies – low base probability may result from an emphasis on novelty and “groundbreaking” research in funding applications, whilst a low grant income per publication may result from low funding rates. The latter, in particular, varies considerably (from ~5% to ~40% across countries, although I am not aware of any studies that have examined the relationship between funding rate and power distribution.

Similarly, certain fields may be more likely to have a high base rate probability than others – clinical trials of medical interventions, for example, represent the culmination of a long process of discovery, experimentation, validation, etc. Indeed, the principle of equipoise should suggest that only 50% of trials should demonstrate a benefit for the intervention over the comparator (which is borne out by empirical work). What would the model predict should be the difference between research of this kind compared with more blue-sky discovery research where the base probability might be considerably lower.

In other words, for these models to be more than useful descriptions they should provide us with insights into how changing incentive structures might shape scientists’ behaviour, and the resulting quality of research. There may be opportunities here to make prospective (perhaps even quantitative) predictions that then could be tested – either as part of this paper, if the author has the resources to do so, or in future studies. The author briefly touches on these issues, but I think the manuscript would be stronger if more specific, concrete, prospective predictions were made.

Reviewer #2: This is a very timely and important article. A few minor revisions could improve the manuscript.

• “IF is a positive constant reflecting mean grant income per publication.” Can we also think about this parameter as a reflection of the “cost to collect data.” If “IF” is large, this translates to the potential profit being higher. Does this imply that the relative cost per sample is small? Conversely, if “IF” is small, does this translate to a situation where data is relatively expensive? Can you discuss how your results reflect on fields where collecting is expensive versus fields where collecting data is relatively inexpensive? On a related note, see Sassenberg et al. (2019) who conclude that a journal’s “demand for higher statistical power [...] evoked strategic responses among researchers. [...] [R]esearchers used less costly means of data collection, namely, more online studies and less effortful measures.''

• Consider exploring (or at the very least commenting) on a nonlinear cost for sample size. In reality, the cost of increasing one’s sample size from 10 to 20 might be different than increasing the sample size from 100 to 110.

• “Which sample sizes are scientifically ideal is of course a complex question in itself, and will depend not only on the cost of sampling but also on the scientific values of true and false positives as well as true and false negatives.” I suggest a comment or a reference on the meta-analytic impact of studies with low power. Consider for example the conclusions of Stanley et al. (2017) in “Finding the power to reduce publication bias.”

• “For instance, Campbell and Gustafson [43] propose conditional equivalence testing to increase the publication value of negative findings.” I suspect that if, in your model, both positive and negative findings were given equal value, the optimal sample size would still be very low. If, regardless of the outcome, the study will be published and the “IF” received, won’t it be optimal to conduct a large number of very small studies? With this in mind, could you elaborate on “increase the publication value of negative findings”? In other words, for the equation “Profit = IF * TPR – s”, what could we consider for replacing the TPR term? How should we be compensating researchers for their work? While I understand that your model is only an approximation of the complicated research economy, can you point to any alternatives to giving researchers a certain amount of grant money per (positive) publication?

Small things:

• “For simplicity we assume they receive funding only, if they publish and they can publish only positive results.” This is an important idea and it needs to be made crystal clear. Please consider rewriting this and perhaps elaborating.

• “To compute the implied distribution of emergent power and positive predictive values, the corresponding values for each ESS were weighted by its TPR/ESS.” This is another very important idea. I suggest writing out an example to make sure the reader understands. For instance: “For example, with the same total amount of ressources at hand, a researcher could conduct 10 small studies with a sample size of 10 or one 2 large studies with sample size of 50, … the emergent studies (i.e., the published literature) will then have….”

• “Two, now prominent, studies from the pharmaceutical industry suggested reproducibility rates of 11 and 22% [1,2, respectively].” Consider adding here a comment/reference to Johnson et al. (2017) “On the Reproducibility of Psychological Science.”

• Fix punctuation and spacing: “chosen for a set of parameters (b, d, IF , see table1)”

6. PLOS authors have the option to publish the peer review history of their article (what does this mean?). If published, this will include your full peer review and any attached files.

Reviewer #1: Yes: Marcus Munafo

Reviewer #2: No

---

## [Author Response · Author response to Decision Letter 0]

24 Jan 2020

Reviewers' comments:

Review Comments to the Author

Reviewer #1: The author presents a relatively simple model, whereby scientists choose “economically rational” sample sizes on the basis of the trade-off between the costs of collecting data (with larger studies being more expensive) and the value of publications (i.e., resulting grant income). The authors acknowledge the relatively simplicity of the model, but suggest that the model performs reasonably well in terms of predicting the distribution of statistical power in a way that mirror empirical efforts to estimate this.

The work is a valuable addition to the literature, and complements previous efforts that have taken a similar approach (i.e., using modelling) to understand the impact of incentive structures on the behaviour of scientists, and the resulting quality of published research. However, I felt that (notwithstanding the virtues of simplicity) it would have been interesting to explore the impact of modifying the input parameters on the predicted outputs of the model (i.e., on scientists’ behaviour).

For example, the drivers of small sample size in the model are low base probability, small effect size, and/or low grant income per publication. One could argue that at least two of these are shaped by funding agencies – low base probability may result from an emphasis on novelty and “groundbreaking” research in funding applications, whilst a low grant income per publication may result from low funding rates. The latter, in particular, varies considerably (from ~5% to ~40% across countries, although I am not aware of any studies that have examined the relationship between funding rate and power distribution.

Similarly, certain fields may be more likely to have a high base rate probability than others – clinical trials of medical interventions, for example, represent the culmination of a long process of discovery, experimentation, validation, etc. Indeed, the principle of equipoise should suggest that only 50% of trials should demonstrate a benefit for the intervention over the comparator (which is borne out by empirical work). What would the model predict should be the difference between research of this kind compared with more blue-sky discovery research where the base probability might be considerably lower.

In other words, for these models to be more than useful descriptions they should provide us with insights into how changing incentive structures might shape scientists’ behaviour, and the resulting quality of research. There may be opportunities here to make prospective (perhaps even quantitative) predictions that then could be tested – either as part of this paper, if the author has the resources to do so, or in future studies. The author briefly touches on these issues, but I think the manuscript would be stronger if more specific, concrete, prospective predictions were made.

Response:

These comments are well taken. I agree fully, that funding agencies have the power to affect IF and b and potentially even d. Indeed, I have attempted to provide exactly the kind of insight about how changing input-parameters would change economically optimal behavior in figures 2 and 3, but have apparently not explained this well. The analysis shown in figure 2 was designed precisely to show the reader how modifying the input parameters individually or in combination will affect the equilibrium sample size and resulting power. These data show clearly that, ceteris paribus, a policy to increase any of these three variables individually (moving down or right across subplots or right on individual x-axes) will tend to promote better powered studies. They further provide a useful tool to glean their interactions. For instance, if my field is interested in effects of d≥0.5 and we assume equipoise (b=0.5), then income per publication should support at least 500 samples (IF=500) to support well powered research (Fig.2B). Importantly, it also implies that small d or b can be, to a certain degree, compensated by a high IF. Finally, it reveals dead spaces (e.g. IF=10), where no plausible b or d lead to power >50%.

In addition to this map of the parameter space I have attempted to show what distributions of output power might emerge, if different distributions of input parameters are assumed (Fig.3). This provides not only insight as to how the most empirically plausible distributions (Fig.3C, c9-c16) would translate to empirically verifiable power distributions but also how other, perhaps more desirable distributions would. Indeed, Fig.3C, c1-c8 shows precisely the case the reviewer asked for, in which 50% of research niches have b>0.5, e.g. because funding agencies dedicate funding specifically to confirmatory research. Similarly, the rows in Fig. 3C show outcomes if income per publication is increased. 

I agree fully, that it would be extremely interesting to explore the empirical relation of funding rates, IF and the resulting power distributions, and am also unaware of research investigating this. Similarly, in line with the reviewer’s suggestion, I strongly suspect that a closer investigation of the empirical literature on power would reveal many high powered studies to be clinical studies (with substantial prior evidence and thus high b). Unfortunately I do not presently have the means to explore this further. However, I have attempted to make the effects of varying input parameters, as deliberated above (Fig.2), as well as the predictions if e.g. funders placed a greater emphasis on confirmatory research (Fig.3) or increased funding ratios (Fig.2, 3) clearer in the manuscript (lines 206, 219ff, 236, 260ff).

I have further added an exploration of a particular policy prescription, namely conditional equivalence testing, as suggested by reviewer 2 (lines 266ff). Together, I believe this offers a range of prospective predictions, both to verify the model and to guide policy.

Reviewer #2: This is a very timely and important article. A few minor revisions could improve the manuscript.

• “IF is a positive constant reflecting mean grant income per publication.” Can we also think about this parameter as a reflection of the “cost to collect data.” If “IF” is large, this translates to the potential profit being higher. Does this imply that the relative cost per sample is small? Conversely, if “IF” is small, does this translate to a situation where data is relatively expensive? Can you discuss how your results reflect on fields where collecting is expensive versus fields where collecting data is relatively inexpensive? On a related note, see Sassenberg et al. (2019) who conclude that a journal’s “demand for higher statistical power [...] evoked strategic responses among researchers. [...] [R]esearchers used less costly means of data collection, namely, more online studies and less effortful measures.''

Response: Thank you for this comment. This is exactly right. As we state in the manuscript (line 68 in the revised manuscript with tracked changes): ‘For simplicity we scale IF as the number of samples purchasable per publication such that the cost of experimentation reduces to sample size (s)’. Accordingly, the predictions and evidence presented by Sassenberg and Ditrich (2019) fits squarely in the model framework presented here. We have included this in the discussion (lines 390ff).

• Consider exploring (or at the very least commenting) on a nonlinear cost for sample size. In reality, the cost of increasing one’s sample size from 10 to 20 might be different than increasing the sample size from 100 to 110.

Response: This is true. Nonlinear cost functions may affect ESS. I have run some exploratory analyses varying the marginal sample cost by including an exponent (cost = svarCost, with s= sample size and varCost= 0.5 to 1.1). While this affected equilibrium sample sizes, the change was mostly due to the resulting change in mean slope (i.e. was similarly and more simply captured by changing IF). In theory, diminishing marginal costs would be relevant particularly if there is a range where sample costs start quickly diminishing, and this is in the range where power approaches 80% or higher (where the Income curve begins to saturate in Fig. 1A). Above this range diminishing marginal cost will have a minor effect, since marginal returns will approach 0 (with 100% Power the publication rate remains the same for larger sample sizes). If marginal cost diminishes quickly below this range, then we can arguable focus on the cost slope above. In other words, for the curvature of the cost function to play a major role for the equilibrium sample size, it would need to be very prominent around exactly the range of sample sizes where the curvature of the power-function is strongest. I would thus cautiously contend that in most cases, a linear cost function (with adjustable slope) is a good approximation. I have now included a brief discussion of this (lines 459ff).

• “Which sample sizes are scientifically ideal is of course a complex question in itself, and will depend not only on the cost of sampling but also on the scientific values of true and false positives as well as true and false negatives.” I suggest a comment or a reference on the meta-analytic impact of studies with low power. Consider for example the conclusions of Stanley et al. (2017) in “Finding the power to reduce publication bias.”

Response: Thank you for this comment. I have modified the manuscript accordingly (line 356).

• “For instance, Campbell and Gustafson [43] propose conditional equivalence testing to increase the publication value of negative findings.” I suspect that if, in your model, both positive and negative findings were given equal value, the optimal sample size would still be very low. If, regardless of the outcome, the study will be published and the “IF” received, won’t it be optimal to conduct a large number of very small studies? With this in mind, could you elaborate on “increase the publication value of negative findings”? In other words, for the equation “Profit = IF * TPR – s”, what could we consider for replacing the TPR term? How should we be compensating researchers for their work? While I understand that your model is only an approximation of the complicated research economy, can you point to any alternatives to giving researchers a certain amount of grant money per (positive) publication?

Response: It is true, that rewarding the publication of each negative finding independent of sample size would promote even lower power (indeed I believe the optimum would consistently be the minimum sample size). This is intuitively nonsensical, because such extreme underpowering will be judged as problematic for the legitimate reason that ‘absence of evidence is not evidence of absence’.

Conditional equivalence testing avoids this problem by separating statistically significant negative findings from inconclusive results. The TPR term then includes statistically significant positive as well as negative results (but not inconclusive results). It thus augments the economic incentive for sufficient power to detect positive results with an economic incentive for sufficient power to detect negative results. This is now explored in the revised manuscript (lines 76ff, lines 266ff, new Fig.4).

Small things:

• “For simplicity we assume they receive funding only, if they publish and they can publish only positive results.” This is an important idea and it needs to be made crystal clear. Please consider rewriting this and perhaps elaborating.

Response: I have added some lines elaborating (lines 59ff)

• “To compute the implied distribution of emergent power and positive predictive values, the corresponding values for each ESS were weighted by its TPR/ESS.” This is another very important idea. I suggest writing out an example to make sure the reader understands. For instance: “For example, with the same total amount of ressources at hand, a researcher could conduct 10 small studies with a sample size of 10 or one 2 large studies with sample size of 50, … the emergent studies (i.e., the published literature) will then have….”

Response: I have added two sentences elaborating (lines 148ff).

• “Two, now prominent, studies from the pharmaceutical industry suggested reproducibility rates of 11 and 22% [1,2, respectively].” Consider adding here a comment/reference to Johnson et al. (2017) “On the Reproducibility of Psychological Science.”

Response: In my understanding, the study by (Johnson et al., 2017) essentially reanalyzes the results from (Open Science Collaboration, 2015), not to judge the reproducibility rates of the published literature, but to infer unknown quantities such as b (base probability of true effects among all tests performed). They find that an observed reproducibility rate of ~36% suggests a b of ~0.1, where ~90% of performed tests never enter the literature. Note, that this is highly consistent with the present findings (Fig. 3 c13-16). Since their study does not add to reproducibility estimates, I would prefer not to cite them in the suggested context. I have however, added a citation where the estimation of b is addressed (line 243, 523).

• Fix punctuation and spacing: “chosen for a set of parameters (b, d, IF , see table1)”

Response: I will go through the document again to check for punctuation and spacing errors. In the shown example (line 103) there is in fact no space after IF and the appearance is due to the latex math environment. In my understanding, according to PloS guidelines, I should use math environment whenever writing variables to make their appearance uniform.

---

## [Decision Letter · Decision Letter 1]

11 Feb 2020

A simple model suggesting economically rational sample-size

choice drives irreproducibility

PONE-D-19-26742R1

Dear Dr. Braganza,

We are pleased to inform you that your manuscript has been judged scientifically suitable for publication and will be formally accepted for publication once it complies with all outstanding technical requirements.

With kind regards,

Luis M. Miller, Ph.D.

Academic Editor

PLOS ONE

Reviewers' comments:

Reviewer's Responses to Questions

**Comments to the Author**

1. If the authors have adequately addressed your comments raised in a previous round of review and you feel that this manuscript is now acceptable for publication, you may indicate that here to bypass the “Comments to the Author” section, enter your conflict of interest statement in the “Confidential to Editor” section, and submit your "Accept" recommendation.

Reviewer #1: All comments have been addressed

Reviewer #2: All comments have been addressed

2. Is the manuscript technically sound, and do the data support the conclusions?

Reviewer #1: Yes

Reviewer #2: Yes

3. Has the statistical analysis been performed appropriately and rigorously? 

Reviewer #1: Yes

Reviewer #2: Yes

4. Have the authors made all data underlying the findings in their manuscript fully available?

Reviewer #1: Yes

Reviewer #2: Yes

5. Is the manuscript presented in an intelligible fashion and written in standard English?

Reviewer #1: Yes

Reviewer #2: Yes

6. Review Comments to the Author

Reviewer #1: The author has addressed all the comments raised in the previous round; I have no further suggestions for improving the manuscript.

Reviewer #2: Two very minor suggestions:

Line 280 – “In the following, we explore the effects of CET given either = 0.5d or = d (Fig. 4).”

You should add a note to clarify that you have alpha=0.05 for both the testing of the null and the testing of equivalence. (Campbell&Gustafson suggest that other options could be considered.) Also, perhaps you might want to add that there is a one-to-one correspondence between the alpha and the width of the equivalence margin (inverse relationship).

Line 486 – “have explored the the effect of various”

Remove second “the”

7. PLOS authors have the option to publish the peer review history of their article (what does this mean?). If published, this will include your full peer review and any attached files.

Reviewer #1: Yes: Marcus Munafo

Reviewer #2: No

---

## [Editor Report · Acceptance letter]

14 Feb 2020

PONE-D-19-26742R1 

A simple model suggesting economically rational sample-size
choice drives irreproducibility 

Dear Dr. Braganza:

I am pleased to inform you that your manuscript has been deemed suitable for publication in PLOS ONE. Congratulations! Your manuscript is now with our production department. 

With kind regards,

on behalf of

Dr. Luis M. Miller 

Academic Editor

PLOS ONE